# Increasing Measurement Agreement Between Different Instruments in Sports Environments: A Jump Height Estimation Case Study

**DOI:** 10.3390/s25175354

**Published:** 2025-08-29

**Authors:** Chiara Carissimo, Annalisa D’Ermo, Angelo Rodio, Cecilia Provenzale, Gianni Cerro, Luigi Fattorini, Tommaso Di Libero

**Affiliations:** 1Department of Medicine and Health Sciences “Vincenzo Tiberio”, University of Molise, 86100 Campobasso, Italy; chiara.carissimo@unimol.it (C.C.); gianni.cerro@unimol.it (G.C.); 2Department of Human, Social and Health Sciences, University of Cassino and Southern Lazio, 03043 Cassino, Italy; annalisa.dermo@unicas.it (A.D.); tommaso.dilibero@unicas.it (T.D.L.); 3Department of Electrical and Information Engineering, University of Cassino and Southern Lazio, 03043 Cassino, Italy; cecilia.provenzale@unicas.it; 4Department of Physiology and Pharmacology “Vittorio Erspamer”, Sapienza University of Rome, 00185 Rome, Italy; luigi.fattorini@uniroma1.it

**Keywords:** measurement comparison, linear regression, convergence, compatibility, assessment test, jump height

## Abstract

The assessment of physical quantity values, especially in case of sports-related activities, is critical to evaluate the performance and fitness level of athletes. In real-world applications, motion analysis tools are often employed to assess motor performance in subjects. In case the methods used to calculate a specific quantity of interest differ from each other, different values may be provided as output. Therefore, there is the need to get a coherent final measurement, giving the possibility to compare results homogeneously, combining the different methodologies used by the instruments. These tools vary in measurement capabilities and the physical principles underlying the measurement procedures. Emerging differences in results could lead to non-uniform evaluation metrics, thus making a fair comparison unpracticable. A possible solution to this problem is provided in this paper by implementing an iterative approach, working on two measurement time series acquired by two different instruments, specifically focused on jump height estimation. In the analyzed case study, two instruments estimate the jump height exploiting two different technologies: the inertial and the vision-based ones. In the first case, the measurement value depends on the movement of the center of gravity during jump activity, while, in the second case, the jump height is derived by estimating the maximum distance ground–foot during the jump action. These approaches clearly could lead to different values, also considering the same jump test, due to their observation point. The developed methodology can provide three different ways out: (i) mapping the inertial values towards the vision-based reference system; (ii) mapping the vision-based values towards the inertial reference system; (iii) determining a comprehensive measurement, incorporating both contributions, thus making measurements comparable in time (performance progression) and space (comparison among subjects), eventually adopting only one of the analyzed instruments and applying the transformation algorithm to get the final measurement value.

## 1. Introduction

Monitoring and analyzing motor abilities are fundamental for improving training efficiency, enhancing performance, and reducing injury risk [1,2]. Key components such as strength, power, endurance, and speed are usually assessed through standardized tests and advanced tools like accelerometers, force platforms, and video analysis systems [3,4,5,6]. These technologies ensure accurate data collection and allow the optimization of training programs by tailoring variables such as volume, intensity, and frequency [7,8].

However, in contexts where access to professional equipment is limited (such as outdoor sports, remote training, or home-based assessments), low-cost and easily replicable methods are preferred [9,10,11]. In these situations, functional field tests or commercial tools based on smartphone cameras and inertial measurement units (IMUs) are often used [12,13]. When analyzing the same motor task with different methods, different aspects can be measured, such as the displacement of the center of mass or the height of the jump [12,13]. While the measured quantities may differ between instruments, these methods can offer valuable insights into an individual’s motor performance when compared against standardized test norms tables [14,15].

To ensure consistency across different methodologies, it is essential to establish correlations and develop procedures that allow meaningful comparison of results obtained with heterogeneous tools [16]. Recent studies have already explored these aspects, especially for vertical jump assessment, comparing force platforms, accelerometers, and smartphone-based applications [17,18,19,20,21].

Nevertheless, such affordable systems often prove to be limited, with proprietary applications that restrict access to the raw data. This could present an obstacle during the data processing and comparison phases when several results are obtained for evaluating the subject’s final performance. Against this backdrop, the present study aims to develop an iterative algorithm working on two measurement time series, obtained by two different instruments, exploiting the above cited technologies. To the best of our knowledge, very few works addressed such an issue, as most of them exploit multiple instrumentation to perform a data fusion scheme, where different information derives from each instrument. In the present case, we need to obtain a map of the same measurement quantity acquired by two instruments. In particular, the aim is twofold: from one side, mapping measurement values of one instrument towards the other one; from the other side, to get a third value, obtained by a processing applied to values acquired by the instruments. Indeed, in case of unavailability of one of the instruments, measurements taken with the other one can be easily reported towards the missing device; furthermore, in case of instruments known to underestimate and overestimate the measurand, respectively, the approach to get a third value can enhance the measurement accuracy. The method is presented considering a specific case study: the measurement of a jump using two different instruments, one based on the inertial measurement principle (Instr1) and the other adopting image-based measurements (Instr2). Instr1 evaluates the jump based on the displacement of the center of gravity during the exercise, while Instr2 calculates the jump based on flight time, i.e., the distance between the floor and the maximum height reached by the feet during the leap.

The sections of this work are organized as follows: Section 2 will describe the motivation for the work, Section 3 the materials and methods adopted, Section 4 the results, and Section 5 and Section 6 the discussion and conclusion, respectively.

## 2. Motivation and Contribution of the Work

Using two different measurement instruments to assess the same physical quantity is a common practice in practical performance assessment contexts, such as field evaluations, or remote/online training sessions. And it is not unusual for different tools to be used throughout the same training or monitoring period in such contexts. While gold-standard instruments do exist and are often used as references in controlled and/or validation studies, access to such equipment is frequently limited due to high costs or low portability. As a result, in real-world settings, practitioners may find themselves comparing values recorded at different times with different devices, which introduces a critical methodological issue: how to evaluate and compare these measurements reliably, especially when none of the tools can be definitively considered the reference. Specifically, concerning the vertical jump assessment, multiple studies have confirmed the reliability of various devices in tracking performance and monitoring workload in high-intensity environments. These devices typically assess the same motor task using two distinct approaches: video-based analysis and IMU-based measurement systems [22,23]. This challenge is further amplified by the growing adoption of hybrid methodologies and settings, which increasingly combine in-person and remote interactions. Such hybrid models require greater flexibility in measurement techniques and, more importantly, necessitate the ability to compare test results collected through different instruments that operate based on fundamentally different physical principles. As a result, the raw outputs of these tools often lack direct comparability, despite referring to the same motor task. The issue that two different instruments provide different measured values and the combined effect of lack of a priori information and difficulty to have repeated measurements compromise both the tracking of performance over time and the interpretation of changes related to training or rehabilitation interventions.

This study addresses these problems by proposing an iterative procedure designed to enhance the agreement between two instruments that concurrently measure the vertical jump height. Specifically, the proposed method uses mutual linear regression applied iteratively to the raw measurement time series, constrained by a shape-preserving criterion, a maximum number of iterations, and a minimum improvement threshold per iteration. Further, a simpler version of the method also allows us to transform the measurement values obtained by one of the instruments in the other one’s reference system through a bi-directional calibration procedure (“Instr1 vs. Instr2” and “Instr2 vs. Instr1”).

The reason why the proposed method can go towards two directions can be described as follows: if none of the involved instruments can be considered a reference, both measurement time series could not be so close to the actual value. Therefore, a third value could reduce the distance between the actual and the measured value, by enhancing the measurement accuracy (in the case explained in the introduction). In the other case, inter-instrument repeatability is improved. In the sports field, to coherently measure the athlete’s performancetime is an important aspect and it should also be robust to measurement instrument change. In this last case, the adoption of a bidirectional alignment between Instr1 and Instr2 is preferable.

A key motivation behind this work is to improve the quality of performance evaluation in remote or resource-limited contexts.

The main contributions of this work are as follows:

The introduction of a reference-independent strategy for improving inter-instrument agreement in performance measurements.The development of an iterative approximation algorithm based on mutual regression that increases the level of agreement between measurements by providing a final, intermediate, and complete measurement as output.The proposal of a generalizable methodological framework applicable to other domains where cross-device comparison is required without access to a gold standard.The emphasis on enhancing the performance of affordable measurement tools to support high-quality assessments in low-cost and decentralized environments.

## 3. Materials and Methods

### 3.1. Participants

A total of 35 male participants (ranging from 28 to 46 years) were recruited for this study. The sample comprised students and teaching staff from the University of Cassino and Southern Lazio. All subjects were physically active according to ACSM’s guidelines [24]. Inclusion criteria were as follows: to be considered as an active subject and to not have any osteoarticular injuries and/or functional limitations that could interfere with the motor task. Recruitment was carried out by sending an email invitation to professors and researchers. Those who expressed interest and responded positively to the invitation were included in the study. The anthropometric characteristics of the sample, as detailed in Table 1, display a broad range of variability due to the sample’s non-normal distribution. All participants were informed about how the protocol would be carried out and provided their consent before participating in the study. Moreover, informed consent and authorization about benefits and risks was obtained in accordance with the Declaration of Helsinki for Human Research of 1964. The study was approved by the Institutional Review Board of the University of Cassino and Southern Lazio no. 24777.2022.12.12).

### 3.2. Instrumentation

This standardized procedure [25] was followed to ensure the correctness of the execution of the squat jump. Measurement data were collected from both the G-Vert (Instr1) sensor (2016, Vert Wearable Fitness Technology, Miami, FL, USA), and the My Jump 2 (Instr2) app (2015, Spain, My Jump 2 Lab). Instr1 is a small inertial sensor (Dimensions of 23 mm × 55 mm × 7.9 mm) that records and calculates the vertical displacement of each jump. It could be synchronized with an app, providing real-time athletic performance monitoring, offering movement data to prevent injuries [26]. Instr2 is a smartphone application that utilizes the camera to measure vertical jump performance. Analyzing jump videos, it calculates key parameters, including jump height, flight-time, and speed, facilitating easy and effective progress tracking. The accuracy and reliability of this instrumentation has also been demonstrated by Brooks et al. [27].

### 3.3. Measurement Protocol of the Case Study

A total of 245 SJ were collected from university students and teaching staff, who were instructed to jump as high as possible following the Squat Jump (SJ) Protocol.

All participants completed a familiarization session and a standardized 10 min warm-up. Each subject performed 7 SJs in sequence, with a rest interval of approximately 10 min between attempts. This recovery period guaranteed full metabolic restoration and matched the organizational flow of the testing sessions.

The following standardized instructions were given to all participants to ensure consistent execution of the SJ:Start with hands firmly placed on the hips.Maintain 90° knee flexion, with the femur parallel to the floor.Perform a maximal vertical jump while keeping hands on the hips and hip–knee extension.Land softly by bending the knees to absorb impact.

Measurements were recorded simultaneously using two different systems: an Inertial Sensor Unit and a video-based system, respectivelly called in this paper Instr1 and Instr2. Both systems provided jump height values in centimeters. Instr1 was securely placed at the L5 vertebra using an elastic belt with an integrated sensor pocket. Simultaneously, the execution of the SJ was recorded by an operator using a smartphone at 60 fps to allow Instr2 to compute flight time. Although the video recording allowed visual verification of execution, a physiotherapy goniometer was used to ensure the correct 90° knee angle before each attempt. Following the Instr2 guidelines, the smartphone was positioned on a tripod at 30 cm from the floor and 150 cm from the participant [28].

### 3.4. Preliminary Analysis

The time series (TS) measured by Instr1 and Instr2 were called TS1 and TS2, respectively. The measured values have been organized in uniform-width classes to create a comparison histogram (Figure 1). It was computed using a probability density function (pdf) option to normalize the measurements. Specifically, the bin value $b_{i}$ was calculated according to the following formula:(1)$$b_{i}=\frac{e_{i}}{\left(N*w_{i}\right)}$$
where $e_{i}$ is the number of elements in the bin, *N* is total number of input elements, and $w_{i}$ is the width of the bin. It is shown by the histogram in Figure 1 that agreement between measurements is already present when the initial dataset is considered. Furthermore, considering a fitting with Gaussian distribution (see Figure 2), a further comparison can be performed and the overlapping area can be computed as an index of agreement. In this case, the value of the overlapping area is 0.76.

Furthermore, the Linear Weighted Cohen’s Kappa statistical coefficient has been computed. Its computation requires a class subdivision of the data (TS1 and TS2).

To obtain a suitable class width, the measurement gap (MG) has been defined. Specifically, it is the distance between the data acquired by the two analyzed instruments as reported in Equation (Equation 2).(2)$${MG}_{i}=\sqrt{\frac{\sum_{i=1}^{n}{\left({TS1}_{i}-{TS2}_{i}\right)}^{2}}{n}}$$
where TS1 and TS2 are the initial measurements acquired by the two instruments, n is the number of observation.

The obtained MG value is 9.15 cm. Based on these results, a class width of 10 cm was chosen for the calculation of the Cohen’s Kappa coefficient.

Accordingly, 5 classes have been hypothesized:

*value* ≥ 10 cm and *value* < 20 cm;*value* ≥ 20 cm and *value* < 30 cm;*value* ≥ 30 cm and *value* < 40 cm;*value* ≥ 40 cm and *value* < 50 cm;*value* ≥ 50 cm and *value* < 60 cm;

The computed value is equal to 0.41 (*p* < 0.001), indicating moderate agreement between the two sets on a scale of 0 to 1 [29].

Based on these initial results, two algorithms have been developed: the first one is an iterative linear regression algorithm, implemented to calculate an intermediate measure that takes into account both methodologies used to derive the jump height; the second one is a convergence algorithm, where the set of measurements acquired with a specific instrument can be mapped against the other instrument’s reference system. This is particularly important in cases where performance reference tables are available only for one of them, but the on-field measurement process has been carried out with the other one.

### 3.5. The Iterative Regression Algorithm

The block diagram showing the main steps of the first algorithm is presented in Figure 3.

To enhance the agreement between the measurements and obtain an intermediate and comprehensive measurement value, an iterative algorithm based on linear regression has been developed. Particularly, the algorithm base is a mapping between two measurement time series, trying to find a proper functional relationship. Several mathematical functions could be chosen to approximate their actual behavior and the suitable function choice actually depends on figures of merit, as the R-square value. The selection of a linear model is based on its superior performance in comparison to other models that were evaluated during a preliminary phase. In particular, an R-square value of 0.98 is obtained, which is higher than the other models tested.

The generalizability of the algorithm was evaluated before selecting the test and training samples. Tests were performed by randomly selecting different portions of the dataset for the training and testing phases. It was observed that the RMSE value calculated between pairs of measurements suffered negligible variations. Specifically, considering 10 random subsets for the testing phase led to an average RMSE of 3.06 cm, with a standard deviation of 0.12 cm. Subsequently, the dataset is divided into two subsets: a number of measurements are used for the training phase of the algorithm (100/245) and the remainder are used for the testing phase (145/245). The pseudocode Algorithm 1 delineates the primary phases of the developed algorithm.
**Algorithm 1** Iterative algorithm based on linear regression1:**Input**: Jump Measurement from Instr1-Instr22:**%Training phase**3:Length of TS = subset of measures is considered for the training phase (TS1 and TS2)4: * *5:Tmean = Mean of TS1 and TS26:Define Peak To Peak Initial Distance (PPID)7:${PPID}_{TS1}$ = MAX(TS1)−MIN(TS1)8:${PPID}_{TS2}$ = MAX(TS2)−MIN(TS2)9:iteration=1;10: * *11:**repeat**12: * *13:   i=iteration14:   $\left[m_{1i},q_{1i}\right]=polyfit\left(TS1,Tmean,1\right)$15:   $\left[m_{2i},q_{2i}\right]=polyfit\left(TS2,Tmean,1\right)$16: * *17:   $TS1_{n}=m_{1i}*TS1+q_{1i}$18:   $TS2_{n}=m_{2i}*TS2+q_{2i}$19: * *20:   $Tmean=mean(TS2_{n},TS1_{n})$21:   ${MG}_{Value}=Distance\left(TS2n,TS1n\right)$22:   ${CohenK}_{Value}=CohenK\left(TS2n,TS1n\right)$23: * *24:   ${NPPD}_{TS1n}$ = $\frac{MAX(TS1n)-MIN(TS1n)}{{PPID}_{TS1}}$25: * *26:   ${NPPD}_{TS2n}$ = $\frac{MAX(TS2n)-MIN(TS2n)}{{PPID}_{TS2}}$27: * *28:   condition_1 = ${MG}_{Value}\left(i\right)<{MG}_{Value}\left(i-1\right)$;29: * *30:   condition_2 = ${CohenK}_{Value}\left(i\right)>{CohenK}_{Value}\left(1\right)$31: * *32:   condition_3 = $({NPPD}_{TS1n}>0.60)\;\&\;$$\left({NPPD}_{TS2n}>0.60\right)$33: * *34:   iteration=iteration+1;35: * *36:**until** condition_1 & condition_2 & condition_337: * *38:**%Test phase**39:Length of TS = subset of measures is considered for the test phase (TS1 and TS2)40: * *41:$TS1_{n}$ = TS1∗$\prod_{i=1}^{x-1}m_{1i}$+$\sum_{i=1}^{x-1}q_{1i}$∗$\prod_{j=i+1}^{x-1}m_{1j}$42: * *43:$TS2_{n}$ = TS2∗$\prod_{i=1}^{x-1}m_{2i}$+$\sum_{i=1}^{x-1}q_{2i}$∗$\prod_{j=i+1}^{x-1}m_{2j}$44:with x=1−number of iteration45:Output: $TS1_{n}$, $TS2_{n}$

The two operational phases of the algorithm will then be presented and explained.

#### 3.5.1. Training Phase

During the training phase, the aim is to run the algorithm several times to obtain a new set of measurements with a higher level of agreement than that calculated in the initial step. After defining the initial TS signals, the average value is calculated between them and the repeat-until loop is enabled. A linear fit is performed for each TS signal with respect to the mean value. The following elements are then derived: the angular coefficient (m) and the intercept (q). These are used to calculate the new measurement values, $TS1_{n}$ and $TS2_{n}$. These pairs are saved in a matrix, which is used as input for the second stage of the algorithm.

The repeat-until loop is iterated until the three conditions are satisfied, with the following computations being performed at each step: (I) the measurement gap between the two different dataset (MG) is calculated, and it is verified that the new set of data has a smaller MG value than the previous step; (II) the linear weighted Cohen’s Kappa is calculated, and it is verified that the value obtained is always greater than that of step 1; (III) the normalized peak-to-peak distance is obtained by calculating it on the new measurements obtained at the i-th step of the algorithm, and it is checked that this parameter never falls below 60% of the minimum and maximum distance obtained between the original timeseries (PPID). The value of 0.6 has been chosen empirically after having tested several solutions among the extreme values (0 and 1), and it resulted in a good compromise between the range preservation and the capability to increase the agreement value of the measurements. The selection of Cohen’s Kappa metric among similar metrics like Bland–Altman plots or Intraclass Correlation Coefficient (ICC) was based on its capacity to provide a quantitative level of agreement between two measurements, taking into account their initial measurement gap (MG). Indeed, the latter value was used to define the class width. Moreover, differently to (ICC), Cohen’s Kappa is not subject to several statistical assumptions (e.g., normality and homogeneous variance) that can limit its applicability and, consequently, the use of the proposed iterative algorithm. As demonstrated in Figure 4, an iteration of the algorithm is presented, with the top plot providing a visualization of the distance between the TS1 and TS2 measurements and their mean value. The bottom plot illustrates the effect of the first iteration of the algorithm, indicating a convergence of the new ${TS1}_{n1}$ and ${TS2}_{n1}$ timeseries downstream of the first linear regression, in comparison to the initial case.

However, should at least one of the three conditions no longer be verified, the repeat-until loop terminates and the second part of the algorithm is executed.

#### 3.5.2. Test Phase

In this phase, the *m* and *q* values calculated during the *x* iterations are considered, as well as the part of the dataset that was not used in the previous step. The remaining 145 samples of the initial TS are the raw data to be considered. To derive the final measurement values, we calculate the following Equation (Equation 3):(3)$$TSk_{n}=TSk*\prod_{i=1}^{x-1}m_{ki}+\sum_{i=1}^{x-1}q_{ki}*\prod_{j=i+1}^{x-1}m_{kj}$$
where *x* represents the number of iterations performed by the algorithm, *k*, in this case, can be either 1 or 2 and identifies the number of timeseries taken in the analysis. The output will be two modified time series, TS1n and TS2n, which represent the final result of the iterative process.

### 3.6. Convergence Algorithm

While the implemented iterative algorithm enables a third measurement to be obtained as a polynomial combination of two different measurements acquired by Instr1 and Instr2, it is also possible to activate a second algorithm that allows TS1 values to converge to TS2 measurements (or vice-versa). The main algorithm steps are represented in Figure 5 and then explained in depth in Algorithm 2: a linear fitting is implemented to define a mathematical transformation function, getting $m_{a}$ and $q_{a}$, which are the angular coefficient and intercept value, respectively.


Two cases are defined:
CASE A: TS1 converging against TS2 reference system (TS12);CASE B: TS2 converging against TS1 reference system (TS21).
**Algorithm 2** Convergence algorithm
1:**Input**: Jump Measurement from Instr1 or Instr2, CASE2:
**%Training phase**
3:Length of TS = subset of measures is considered for the training phase (TS1 and TS2)4: * *5:%Create a linear fitting:6:

$\left[m_{a},q_{a}\right]=linear\text{\_}fitting\left(TS1,TS2\right)$

7: * *8:
**%Test phase:**
9:Length of TS = subset of measures is considered for the test phase ($TS1_{Test}$ and $TS2_{Test}$)10: * *11:**if** CASE == A **then**12:   TS12=$m_{a}*TS1_{Test}+$$q_{a}$13:   Output: TS1214:
**else**
15:   TS21=$\frac{1}{m_{a}}*TS2_{Test}-$$q_{a}/m_{a}$16:   Output: TS2117:
**end if**




## 4. Results

The following section is concerned with a detailed description of the results obtained. These results are shown to illustrate the outcomes of both the training phase and the final test phase.

### 4.1. Algorithm 1: Training Phase Outcomes

In the presented case study, the iterative algorithm performed six iterations. Table 2 shows all the conditions evaluated at each stage of the algorithm. Notably, the condition that was not exceeded was the normalized amplitude value, which falls to 0.58 below the threshold value in the case of TS1. Iteration 0 represents the initial point at which the data has not been processed yet. The MG is reduced by 38% (from 9.15 cm to 5.65 cm) at step 1 compared to the initial value, and by a further 7% at the following steps, resulting in a final MG value of 3.79 cm. A reduction in error of 58% (from 9.15 cm to 3.79 cm) is observed when the last MG result is compared with the initial one. Cohen’s Kappa value starts at 0.41 in step 0 and increases to 0.72 in step 6, moving from the “moderate agreement” range (0.41–0.60) to the “substantial agreement” range (0.61–0.80) [29]. This statistical result shows that the agreement between the two measures is not accidental (*p* < 0.001).

Before moving on to the second stage of the algorithm at the end of the while loop, the resulting pair of measurements are plotted in a histogram, as shown in Figure 6. It can be seen that there is greater agreement between the number of occurrences per class than in Figure 1. This result is also emphasized by the plots shown in Figure 7, which illustrate how the common area increased from 0.76 before processing (Figure 2) to 0.99 after processing.

### 4.2. Algorithm 1: Test Phase Outcomes

During the testing phase, any part of the initial dataset that was not used in the previous phase is considered raw data. For simplicity, this data is referred to as TS1 and TS2. Applying Equation (Equation 3) to these vectors yields the final measurements, named TS1n and TS2n. Figure 8 shows a histogram comparing the occurrences of the different classes in the raw data. The same is done in Figure 9, where the histogram compares the classes in the TS1n and TS2n measurements.

In both figures, each class was numbered progressively from 1 to 8. Then, mismatches were assessed for each pair of measures—that is, the difference in occurrences for each class, according to the following equation:(4)$$MismatchLevel=\frac{|{OccTS1}_{i}-{OccTS2}_{i}|}{{OccTS1}_{i}+{OccTS2}_{i}}$$
where ${OccTS1}_{i}$ is the number of values that fall into class *i* considering the TS1 measurement; ${OccTS2}_{i}$ is the number of values belonging to class i considering the TS2 measurement, and i is the number of classes, which in this case ranges from 1 to 8.

The results are shown in Table 3. It can be seen that applying the polynomial reduces the mismatch between the new measurements compared to the raw data.

The increase in agreement between the two measurements is also evident from the Gaussians shown in Figure 10 and Figure 11.

Notably, the common area has increased from 0.60 to 0.85, representing a 42% rise. Evaluating the Cohen’s Kappa coefficient reveals that, for the raw data pair, a very low value of 0.23 (*p* < 0.05) is obtained; however, after applying the algorithm, a value of 0.55 (*p* < 0.001) is carried out. Evaluating the ranges described in the paper [29], there is a shift from fair (0.21–0.40) to moderate (0.41–0.60) agreement.

**Figure 10 sensors-25-05354-f010:**
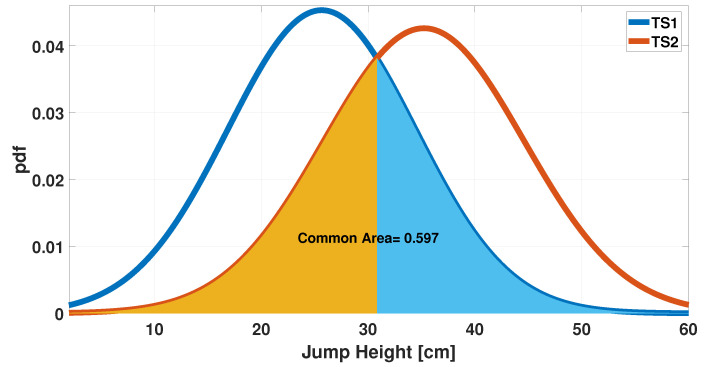
Gaussian distributions of jump height measurements: evaluation of common areas between the raw test data.

**Figure 11 sensors-25-05354-f011:**
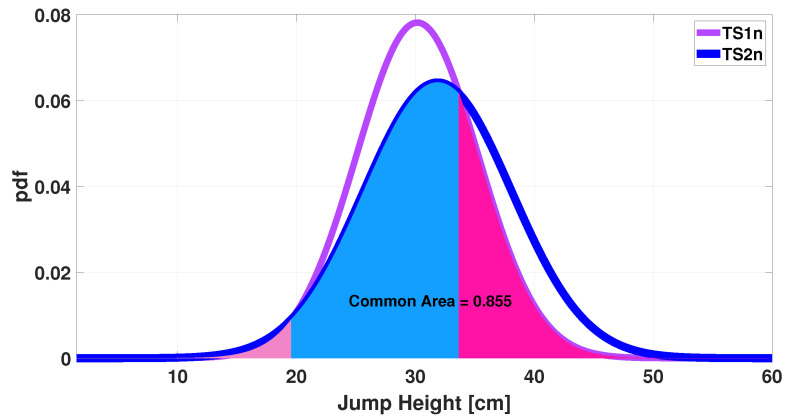
Evaluation of common areas between Gaussian distributions of jump height measurements obtained as a final result of the iterative algorithm.

Finally, further improvements can be seen in the reduction of the distance between the two time series. In particular, the MG values of the TS1 and TS2 time series decrease from 10.68 cm to 3.68 cm after applying the linear polynomial. The results are also shown in Table 4.

### 4.3. Algorithm 2: Training and Testing Outcomes

The second rule of the algorithm is the possibility to transform the measurement obtained by one device in the other one’s system. To evaluate the algorithm performance the two metrics are considered: Root Mean Square Error (RMSE) and Cohen’s Kappa Value. In Table 5 the first row shows the results obtained considering data before the test phase, the distance between the two measures is 10.76 cm, and a fair level of agreement with a Cohen’s Kappa coefficient equal to 0.23 is obtained.

Depending on the input chosen for the algorithm, two scenarios are possible: CASE A and CASE B, which are explained in the previous Section 3. As can be seen in the last two rows of the Table 5, the RMSE value falls below 6 cm in both cases and the level of agreement changes from ‘fair’ to ‘substantial’. These results are also demonstrated in the Figure 12, in which the two analysed cases are plotted. The top graph shows the application of the algorithm when the measurements of Instr2 are referred to Instr1. The second figure illustrates the convergence of the measurements of Instr1 towards those of Instr2. In both cases, the green dotted line represents the convergence result. A summary of the obtained results is reported in Table 5.

## 5. Discussion

Many different measuring instruments are used to assess jump heights in sporting environments. The most commonly used technologies for evaluating jump height are those based on inertial measurements [30,31] and those involving cameras [32,33].

In recent years, there has been a growing tendency to adopt remote or hybrid assessment strategies. These approaches have spread due to their logistical simplicity and cost-effectiveness. For example, students and amateur individuals often participate in physical assessments conducted outside controlled laboratory environments, relying on online platforms or mixed in-person/remote formats to facilitate measurements without requiring travel or access to specialized facilities. In such contexts, it is common for different tools to be used throughout the year or training cycle, either due to changes in available equipment or the rotation of operators responsible for administering the assessments [30,34]. For example, once the magnitude to be measured has been established, it may be useful to use cameras where the athlete cannot wear instrumentation, or wearable inertial systems where cameras cannot be used.

This variability introduces challenges in ensuring data consistency, particularly when the measure is evaluated using different technologies. For instance, jump height may be measured either via center of mass displacement (as in IMU-based systems) or through flight time (as in video-based analysis), which can yield different values. The proposed methodology allows the measurements of Instr1 to be mapped with respect to Instr2 (and vice versa), while also enabling a measurement to be obtained that integrates the contributions of both the inertial (Instr1) and vision-based (Instr2) instruments. This makes it possible to take comparable measurements over time and between different athletes, and ultimately allows the use of a single instrument with a transformation algorithm applied to obtain the final measurement. For this reason two different algorithms were developed and presented. For instance, this approach could benefit athletes and coaches who frequently travel for competitions, as it would allow greater flexibility in planning and managing their trips. Moreover, in other contexts, this approach could enable people such as ‘digital nomads’ to lead a more flexible lifestyle and travel for work or pleasure throughout the year, while maintaining a consistent training routine. Furthermore, limited technical expertise is sufficient to apply the methodology as it could easily be stored in a mobile application where the only option to choose is which Algorithm should be run (1 or 2) and, in the case of Algorithm 2, in which direction it should work (case A or case B).

In order to test the effectiveness of the proposed algorithms, a total of 245 height jump measurements were considered. A number of 100 measurements were used to train the algorithm. Considering Algorithm 1, the results of the training data show a reduction in the MG of the 58% between the first and last stage of the process. Moreover, it is possible to observe an increase in the level of measurement accordance from “moderate” (0.41, *p* < 0.001) to “substantial” agreement (0.61, *p* < 0.001). The results obtained in the testing phase of the algorithm show that when a data set not used for validating the iterative algorithm is used and the linear polynomial is applied, a new pair of data is obtained that shows a 67% decrease in DM compared to the initial TS. Cohen’s Kappa statistical coefficient highlights that the algorithm’s effect leads to a significant moderate agreement (*p* < 0.001) between the pairs of measurements. These results show the effectiveness of the proposed algorithm in reducing the distance between the two measurements towards an intermediate measure that considers both jump height definitions. Furthermore, the results of Algorithm 2 demonstrated how convergence with respect to one of the instruments guarantees a reduction in RMSE of approximately 45% in both cases, also achieving an improvement in the agreement between the measurements, which went from a fair level to a substantial one.

Moreover, current reference tables for functional evaluation are typically based on two main parameters: jump height and relative calculated jump power output in watts. This is because the gold standard for measuring jump performance is the force platform, which estimates jump height by means of the ground reaction forces. Given this standard, a number of validated equations have been developed to estimate power output starting from jump height [35]. Consequently, video-based methods, which also rely on flight time to estimate height, are generally considered to be the closest alternative to force platforms in terms of measurement logic [36].

Low-cost video-based systems often face limitations in real-world or non-elite contexts, being highly affected by factors such as frame rate, occlusions, camera angle, and image quality. The data processing approach proposed in this study offers a way to normalize vertical jump measurements, enabling a more reliable estimation of power output. This allows operators to obtain comparable and meaningful performance metrics even in decentralized or amateur settings, while maintaining consistency with established reference standards. The main objective of this study was to develop and validate an algorithm that could improve concordance between two measurements taken using different instruments in sports environments. In the initial phase of the study, 35 males aged between 28 and 46 were considered. This represents a limitation of the study, and future developments will address this by increasing the size of the dataset, including females, and extending the age range.

## 6. Conclusions

When measuring a specific quantity, the use of instruments based on different technologies can yield results that are not directly comparable. The proposed methodology, which integrates two algorithms (an iterative linear regression and a convergence function), addresses this issue by generating a third value, taking into account contributions from measurements performed by the two involved instruments. This output can either be obtained as a combination of the two initial measurements (based on a linear regression) or as a value that converges towards one of them, depending on the operator’s needs. Our findings show that the application of these algorithms leads to a very low variability when transitioning from one measurement to another, thus improving agreement and comparability between different tools. This characteristic makes the approach particularly valuable in contexts relying on low-cost instruments or heterogeneous testing protocols. Moreover, since the algorithm is fully automated, it does not require advanced technical expertise or professional-grade equipment. As a result, it enables greater flexibility in managing training programs, both in-person and remotely, while ensuring reliable and consistent monitoring of athlete performance. This could allow a better consistency in in-person/remote mixed contexts, allowing a more flexible managing of travelling and working life.

## Figures and Tables

**Figure 1 sensors-25-05354-f001:**
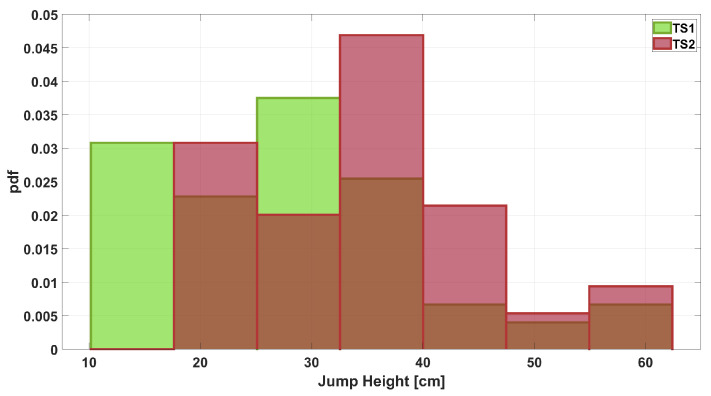
Histogram of the jump height measurements acquired from Instr1 and Instr2.

**Figure 2 sensors-25-05354-f002:**
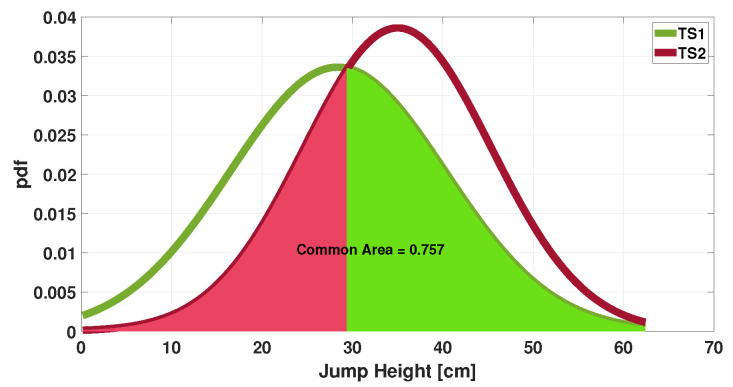
Gaussian distributions of jump height measurements (TS1, TS2): evaluation of common areas.

**Figure 3 sensors-25-05354-f003:**
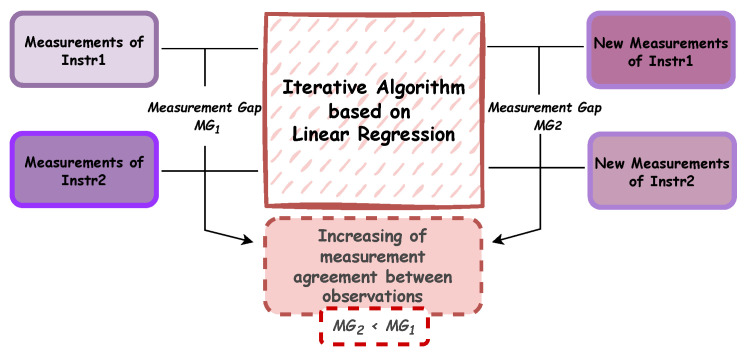
The block diagram shows the main steps of the logical procedure used to develop the iterative linear regression algorithm.

**Figure 4 sensors-25-05354-f004:**
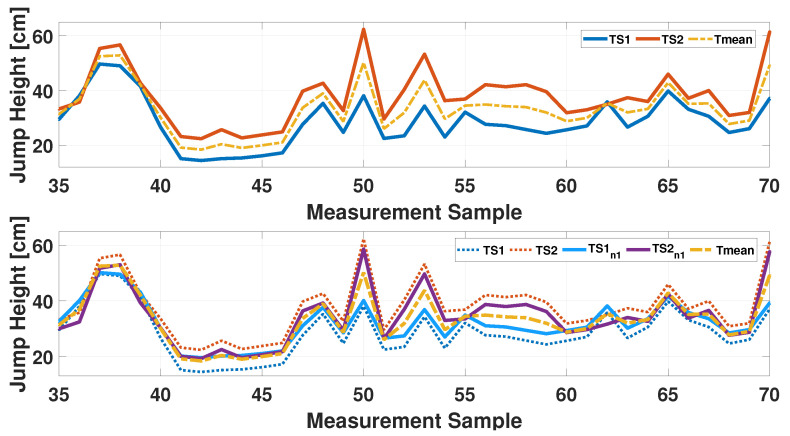
First iteration of the developed algorithm: the top figure shows the TS1 (blue line) and TS2 (orange line) measurement compared with their mean value (yellow line); the bottom figure adds the new data (continuous lines) as a results of the first algorithm iteration.

**Figure 5 sensors-25-05354-f005:**
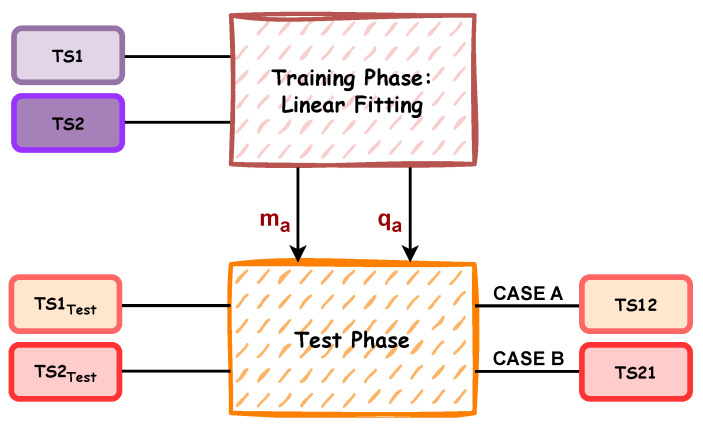
Block diagram of the convergence algorithm divided in two operative phase.

**Figure 6 sensors-25-05354-f006:**
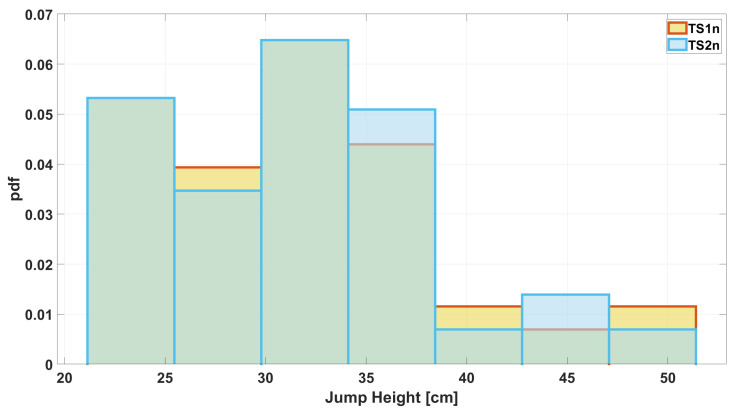
Histogram of the jump height measurements computed at the end of the training phase in the iterative algorithm.

**Figure 7 sensors-25-05354-f007:**
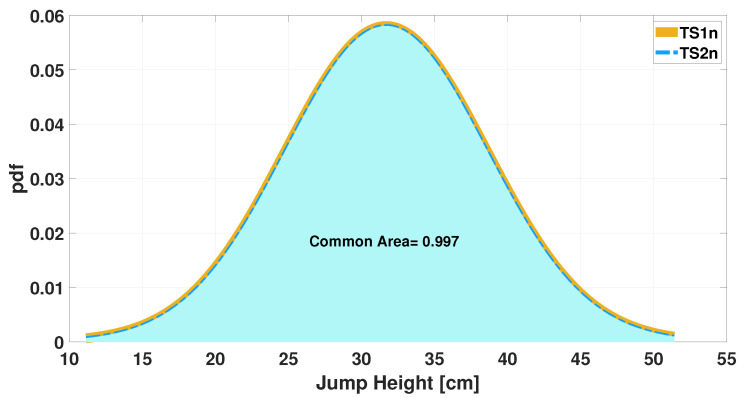
Gaussian distributions of jump height measurements: evaluation of common areas after the end of training phase in the iterative algorithm.

**Figure 8 sensors-25-05354-f008:**
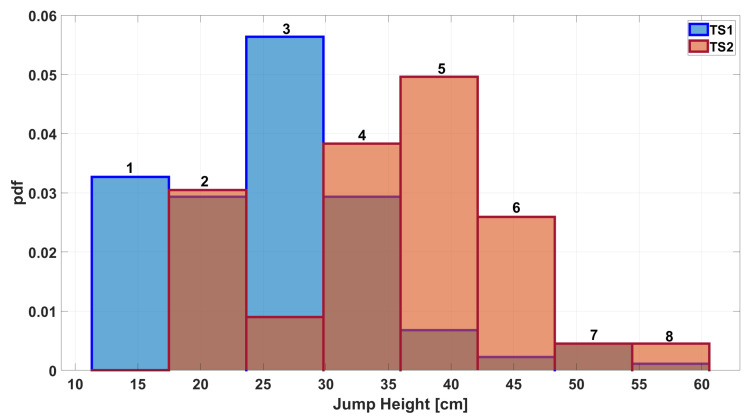
Histogram of jump height measurements when considering raw data during the test phase.

**Figure 9 sensors-25-05354-f009:**
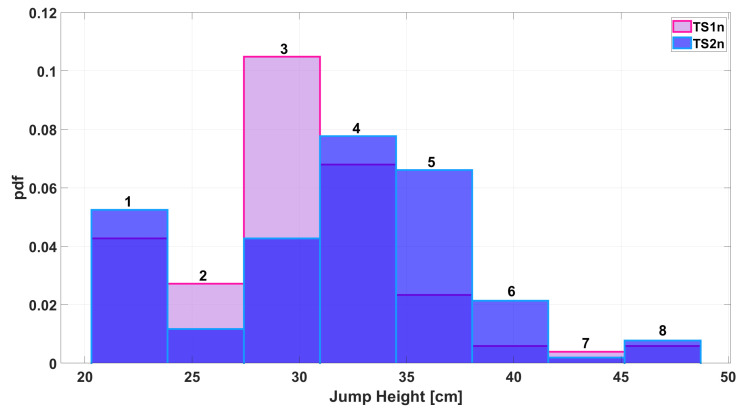
Histogram of jump height measurements when considering final outcomes of the algorithm process (TS1n–TS2n).

**Figure 12 sensors-25-05354-f012:**
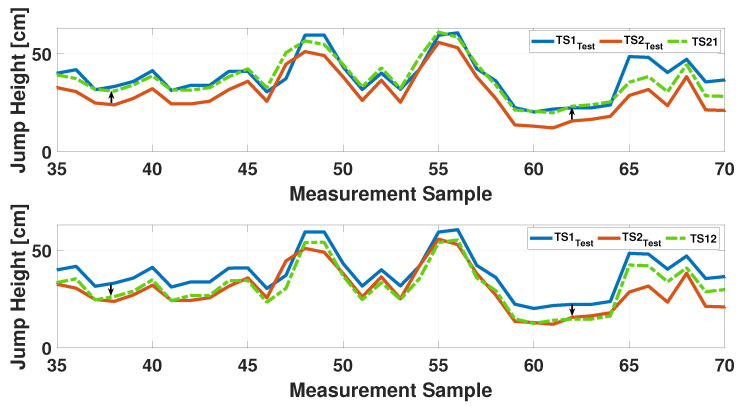
Results of the convergence algorithm: the top figure represents the convergence from TS2 to TS1 in CASE A; the second plot represents the convergence from TS1 to TS2 in CASE B.

**Table 1 sensors-25-05354-t001:** Anthropometric characteristics of total sample. BM: Body Mass; BMI: Body Mass Index.

		*Shapiro–Wilk*	
	Mean	SD	*p*
Age (y)	37.6	8.83	<0.001
Height (cm)	175.7	6.69	0.004
BM (kg)	77.0	11.69	<0.001
BMI (kg/$m^{2}$)	24.9	3.31	<0.001

**Table 2 sensors-25-05354-t002:** Values of the conditions computed for each algorithm iterations.

Algorithm Conditions			Algorithm Iterations						
			0	1	2	3	4	5	6
**MG [cm]**			9.15	5.65	5.22	4.82	4.44	4.12	3.79
**Cohen’s Kappa**			0.41	0.70	0.67	0.61	0.60	0.68	0.72
**Normalized** **Pk-to-Pk Distance**		**TS1**	1	0.87	0.80	0.74	0.70	0.63	**0.58**
		**TS2**	1	0.99	0.91	0.85	0.78	0.72	0.66

**Table 3 sensors-25-05354-t003:** Mismatch level.

	1	2	3	4	5	6	7	8
**Mismatch (TS1, TS2**)	1	0.019	0.72	0.13	0.76	0.84	-	0.60
**Mismatch (TS1n, TS2n**)	0.10	0.40	0.42	0.06	0.48	0.57	0.33	0.14

**Table 4 sensors-25-05354-t004:** Summarized results for Algorithm 1.

	MG [cm]	Cohen’s Kappa Value	Agreement Level
**Before applying the algorithm**	10.68	0.23	Fair
**After applying the algorithm**	3.68	0.55	Moderate
**Percentage of improvement**	(−)64.55	(+)139.13	—

**Table 5 sensors-25-05354-t005:** Summarized results for Algorithm 2.

	RMSE [cm]	Cohen’s Kappa Value	Agreement Level
**Before applying the algorithm**	10.76	0.23	Fair
**After applying the algorithm: CASE A**	5.89	0.64	Substantial
**After applying the algorithm: CASE B**	5.56	0.57	Substantial
**Percentage of improvement: CASE A**	(−)45.26	(+)178.26	—
**Percentage of improvement: CASE B**	(−)48.33	(+)147.83	—

## Data Availability

The data presented in this study are available on request from the corresponding author due to privacy restrictions related to individual participants.

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
