# Peer review of "Increasing Measurement Agreement Between Different Instruments in Sports Environments: A Jump Height Estimation Case Study"

_sensors, 2025, doi:10.3390/s25175354_

Round 1
Reviewer 1 Report
Comments and Suggestions for Authors
- The literature review covers many references, but there is limited critical comparison of the proposed method with existing sensor fusion or cross-device mapping methods. Are there previous attempts to align measurements from inertial and vision-based tools using regression or machine learning? Please provide a more direct comparison.
- While the iterative algorithm is shown to reduce the measurement gap and increase Cohen’s Kappa, it is not entirely clear why producing a third set of values (i.e., a merged time series) is preferable over directly mapping one instrument’s measurements onto the other. Please explain the rationale and potential applications for the intermediate TS1n/TS2n outputs more explicitly (e.g., in training monitoring, which values should be interpreted and why?).
- The cohort consists of male university staff aged 28–46, with relatively wide variability in age and body mass index. Were any considerations given to physical fitness or baseline motor ability? Also, the reason for choosing a 10-minute rest between 7 jumps per participant is not fully justified; most literature prescribes 1–2 minutes. Please elaborate.
- While Cohen’s Kappa and the Gaussian overlap metric are applied throughout the manuscript, their statistical interpretation could benefit from a clearer explanation. For example, what does the common area metric mean practically in this context? Why is it chosen over other similarity metrics such as Bland-Altman plots or Intraclass Correlation Coefficients (ICCs)? Also, define the 0.60 threshold in normalized Pk-to-Pk distance more precisely—why this value?
- While linear models are easier to interpret, real-world discrepancies between sensor systems may involve nonlinearities (e.g., delay, drift, saturation). Have the authors considered polynomial or nonlinear regression models? If so, why were they not chosen? A brief discussion of these modeling choices would be helpful.
- The iterative model is trained on 100 of the 245 samples. Did the authors verify that the model’s performance generalizes beyond this specific subset (e.g., via cross-validation)? Consider including confidence intervals for performance metrics (e.g., RMSE, Kappa) in the test phase.
- The iterative algorithm yields substantial improvement in agreement, but the reader must flip back and forth between figures and narrative text. It would be helpful to include a consolidated comparison table summarizing: initial MG and Cohen’s K values, final values after iteration, % improvement, and number of iterations until convergence, etc.
Author Response
We thank the reviewers for revising the work so carefully, making a great contribution to the work. The reviewed manuscript has now been uploaded, and you can find the corrections in red throughout the text.
- The literature review covers many references, but there is limited critical comparison of the proposed method with existing sensor fusion or cross-device mapping methods. Are there previous attempts to align measurements from inertial and vision-based tools using regression or machine learning? Please provide a more direct comparison.
Response 1: Following the comment, the authors revised the introduction section. To the best of our knowledge, most literature focuses on fusing data deriving from different instruments to catch enhanced information which would be not available in case of one only instrument adoption. Our approach is slightly different, as we intended to map the “same” quantity acquired by two instruments towards a common value that could be in one of the reference systems (of the involved instruments (Instr1 towards Instr2, Instr2 towards Instr1) or a third case getting an agreed value, aiming at enhancing measurement repeatability and, in specific cases, measurement accuracy. We emphasized this concept in the introduction section.
- While the iterative algorithm is shown to reduce the measurement gap and increase Cohen’s Kappa, it is not entirely clear why producing a third set of values (i.e., a merged time series) is preferable over directly mapping one instrument’s measurements onto the other. Please explain the rationale and potential applications for the intermediate TS1n/TS2n outputs more explicitly (e.g., in training monitoring, which values should be interpreted and why?).
Response 2: The response depends on the application field. Particularly, if none of the involved instruments can be considered a reference, both measurement time series could not be so close to the actual value. Therefore, a third value could reduce the distance between the actual and the measured value, by enhancing the measurement accuracy (in case underestimation and overestimation are known phenomena for the involved instruments). In the other case, inter-instrument repeatability is improved.
In the sports field, as explained in the paper, there is also another need: measure coherently the athlete’s performance during time, even if the measurement instrument could change. In this last case, the adoption of a bidirectional alignment between Instr1 and Instr2 is preferable.
Some of the presented considerations have now been added in the paper, specifically in “Motivation and Contribution of the Work”.
- The cohort consists of male university staff aged 28–46, with relatively wide variability in age and body mass index. Were any considerations given to physical fitness or baseline motor ability? Also, the reason for choosing a 10-minute rest between 7 jumps per participant is not fully justified; most literature prescribes 1–2 minutes. Please elaborate.
Response 3: We thank the reviewer for this valuable comment. We have now added details regarding the participants’ physical fitness and baseline motor ability in the “Participants”(“All subjects were physically active according to ACSM's guidelines”).
Regarding the 10-minute rest interval, our choice was based on two main reasons. First, from a physiological standpoint, we aimed to ensure complete resynthesis of creatine phosphate stores to allow each jump to be performed at maximal intensity, minimizing fatigue effects. Second, for practical and logistical reasons, the relatively large sample size required participant rotation, and the average waiting time between turns was approximately 10 minutes. Therefore, this recovery period both guaranteed full metabolic restoration and matched the organizational flow of the testing sessions. We added the following sentence to the Measurement Protocol of the case study section to clarify this: “This recovery period guaranteed full metabolic restoration and matched the organizational flow of the testing sessions.”
- While Cohen’s Kappa and the Gaussian overlap metric are applied throughout the manuscript, their statistical interpretation could benefit from a clearer explanation. For example, what does the common area metric mean practically in this context? Why is it chosen over other similarity metrics such as Bland-Altman plots or Intraclass Correlation Coefficients (ICCs)? Also, define the 0.60 threshold in normalized Pk-to-Pk distance more precisely—why this value?
Response 4: Thanks to the reviewer for the comment. To provide the readers with an intuitive representation of the measurement agreement, the Gaussian overlap metric was proposed. Indeed, it allows both a graphical representation of the measurement distribution and a quantitative index of agreement (i.e., the overlapping area). Cohen’s Kappa allows us to compute the level of agreement between the two measurements, taking into account their initial measurement gap (MG). Indeed, the class subdivision of the data was performed considering the MG value for a suitable class width. Bland-Altman plots provide a graphical method to assess the measurement agreement, but differently from Cohen’s Kappa, it does not provide a final quantitative value. This limit would not have allowed us to assess the changing of agreement in the iterative process. Intraclass Correlation Coefficient (ICC) is subject to several statistical assumptions (e.g., normality and homogeneous variance) that could limit its applicability and, consequently, the use of the proposed iterative algorithm. In subsection 3.5.1, the following sentence was added to better explain the choice made: “The selection of Cohen’s Kappa metric among similar metrics like Bland-Altman plots or Intraclass Correlation Coefficient (ICC), was based on its capacity to provide a quantitative level of agreement between two measurements, taking into account their initial measurement gap (MG). Indeed, the latter value was used to define the class width. Moreover, differently to (ICC), Cohen’s Kappa is not subject to several statistical assumptions (e.g., normality and homogeneous variance) that can limit its applicability and, consequently, the use of the proposed iterative algorithm.”
Finally, the normalized Pk-to-Pk threshold has been chosen following a criterion of measurement range keeping: indeed, as the number of iterations increases, each measurement time series excursion (max - min) tends to decrease, due to the requirement to get closer to the other one. So, without setting some threshold value, the final output would be a flat time series where measurements agree but they become too far from their initial values and excursions. To keep the range pretty unchanged and increase the agreement level of the measurements, we tested a normalized Pk-to-Pk threshold as one of the stop conditions. Clearly, setting it to 1 would have meant to strongly limit the convergence algorithm; on the other hand, setting it to 0 would have led to a completely flat measurement time series effect. We tested several solutions among these extreme values and found 0.6 as a good compromise between the range preservation and the capability to increase the agreement value of the measurements.
- While linear models are easier to interpret, real-world discrepancies between sensor systems may involve nonlinearities (e.g., delay, drift, saturation). Have the authors considered polynomial or nonlinear regression models? If so, why were they not chosen? A brief discussion of these modeling choices would be helpful.
Response 5: Thanks to the reviewer for the comment. In the initial phase of the analysis, several polynomial models were evaluated. In particular, it was observed that a linear function better described the relationship between the two measurement instruments. In this specific case, an R-square of 98% was obtained, while lower values were achieved with polynomials of a higher degree. Based on these preliminary results, an algorithm based on linear regression was implemented. Moreover, the use of a linear regressor over a nonlinear one offers superior computational efficiency in terms of calculation speed and reduced computational cost. These aspects, for example, can be crucial should the algorithm be implemented on a microcontroller. In subsection 3.5, the following sentence was added in order to better justify the choice made: “The selection of a linear model is based on its superior performance in comparison to other models that were evaluated during a preliminary phase. In particular, an R-square value of 0.98 is obtained, which is higher than the other models tested.”
- The iterative model is trained on 100 of the 245 samples. Did the authors verify that the model’s performance generalizes beyond this specific subset (e.g., via cross-validation)? Consider including confidence intervals for performance metrics (e.g., RMSE, Kappa) in the test phase.
Response 6: Thanks to the reviewer for the comment. To verify the generalisability of the model, several tests were conducted using random samples taken from the main dataset in both the training and testing phases. No substantial differences in the algorithm's performance were found in either case. For the testing phase in particular, 10 different random subsets of the initial dataset were considered. The RMSE value was calculated between the pre-analysis test values and the values after application of the algorithm. It can be seen that the average RMSE value in these cases is 3.06 cm, with a standard deviation of 0.12 cm. This result demonstrates the generalisability of the algorithm and its independence from the input data. To better explain this point, the following sentence has been added to the subsection 3.5: “The generalisability of the algorithm was evaluated before selecting the test and training samples. Tests were performed by randomly selecting different portions of the dataset for the training and testing phases. It was observed that the RMSE value calculated between pairs of measurements suffered negligible variations. Specifically, considering 10 random subsets for the testing phase led to an average RMSE of 3.06 cm, with a standard deviation of 0.12 cm. ”.
- The iterative algorithm yields substantial improvement in agreement, but the reader must flip back and forth between figures and narrative text. It would be helpful to include a consolidated comparison table summarizing: initial MG and Cohen’s K values, final values after iteration, % improvement, and number of iterations until convergence, etc.
Response 7: Thanks for the comment. Now tables IV and V report the summary of the obtained results in the test phase, except for the number of iterations, that is needed only in the training phase to get final regression parameters, and it is therefore reported in Table II. In test phase, a single application of the parameters is sufficient, making the method not iterative.
Reviewer 2 Report
Comments and Suggestions for Authors
The article addresses the critical issue of measurement agreement between different instruments in sports environments, focusing on jump height estimation. It presents a novel approach using iterative algorithms to enhance agreement between measurements from inertial and vision-based systems. The study is well-structured, and the methodology is sound. The results demonstrate the effectiveness of the proposed algorithms, which have practical implications for sports performance assessment. However, there are several areas where minor revisions could improve the clarity and impact of the manuscript.
1, While the algorithms are described in detail, the explanation could be made more accessible to readers without a strong technical background. Consider adding a brief summary of the key principles behind the iterative regression and convergence algorithms before diving into the technical details.
2,The introduction mentions previous studies comparing different measurement methods for jump height. However, a more detailed comparison of the proposed algorithms with existing methods for improving measurement agreement would strengthen the manuscript. This could include a discussion of the advantages and limitations of the proposed approach relative to other techniques.
3,The discussion highlights the potential applications of the algorithms in low-cost and decentralized settings. It would be beneficial to provide more specific examples of how the algorithms could be implemented in real-world scenarios, such as by coaches or athletes with limited technical expertise.
4,A more thorough discussion of the limitations of the study would enhance the credibility of the work. For example, considerations about the generalizability of the findings to other types of sports activities or instruments could be addressed.
5,Ensure consistency in the formatting of equations and figures throughout the manuscript. Some sentences are overly complex and could be simplified for improved readability. For instance, in the abstract, "Such tools have different measurement capabilities as well as physical principles over which the measurement procedure is based" could be rephrased as "These tools vary in measurement capabilities and the physical principles underlying the measurement procedures."
Author Response
We thank the reviewers for revising the work so carefully, making a great contribution to the work. The reviewed manuscript has now been uploaded, and you can find the corrections in red throughout the text.
The article addresses the critical issue of measurement agreement between different instruments in sports environments, focusing on jump height estimation. It presents a novel approach using iterative algorithms to enhance agreement between measurements from inertial and vision-based systems. The study is well-structured, and the methodology is sound. The results demonstrate the effectiveness of the proposed algorithms, which have practical implications for sports performance assessment. However, there are several areas where minor revisions could improve the clarity and impact of the manuscript.
1, While the algorithms are described in detail, the explanation could be made more accessible to readers without a strong technical background. Consider adding a brief summary of the key principles behind the iterative regression and convergence algorithms before diving into the technical details.
Response 1: Thanks for the comment. More details about the algorithm, specifically concerning the conditions for iteration stopping ( subsubsection 3.5.1), and information about the regression (subsection 3.5) have now been added to the paper.
2,The introduction mentions previous studies comparing different measurement methods for jump height. However, a more detailed comparison of the proposed algorithms with existing methods for improving measurement agreement would strengthen the manuscript. This could include a discussion of the advantages and limitations of the proposed approach relative to other techniques.
Response 2: To the best of our knowledge, most of the existing literature focuses on combining data from different instruments to extract additional information that would not be accessible using a single device. In contrast, our approach is designed to map the “same” quantity measured by two instruments onto a common value, which can correspond either to one of the reference systems of the devices involved or to a third, harmonized value representing an agreed measurement. These aspects have been focused in the “Introduction” and “Motivation and Contribution of the work” sections.
3,The discussion highlights the potential applications of the algorithms in low-cost and decentralized settings. It would be beneficial to provide more specific examples of how the algorithms could be implemented in real-world scenarios, such as by coaches or athletes with limited technical expertise.
Response 3: Thank you to the reviewer for this insightful suggestion. To better clarify the potential practical application, this sentence has been added in the Discussion session: “For instance, this approach could benefit athletes and coaches who frequently travel for competitions, as it would allow greater flexibility in planning and managing their trips. Moreover, in other contexts, this approach could enable people such as 'digital nomads' to lead a more flexible lifestyle and travel for work or pleasure throughout the year, while maintaining a consistent training routine. Furthermore, limited technical expertise is sufficient to apply the methodology as it could easily be stored in a mobile application where the only option to choose is which Algorithm should be run (1 or 2) and, in case of Algorithm 2, in which direction it should work (case A or case B)”.
4,A more thorough discussion of the limitations of the study would enhance the credibility of the work. For example, considerations about the generalizability of the findings to other types of sports activities or instruments could be addressed.
Response 4: Thank you to the reviewer for the comment. The aim of this preliminary study is to develop and test the aforementioned algorithm. In order to achieve this goal, the study only considered a limited number of male subjects. This is a limitation of the study, as it would be interesting to see what happens when subjects of both genders and different ages are considered. Future developments of this study could address this issue and verify the algorithm's validity and performance as the number of subjects increases, the age range expands and both men and women are included. In the Discussion Section more information is added: “The main objective of this study was to develop and validate an algorithm that could improve concordance between two measurements taken using different instruments in sports environments. In the initial phase of the study, 35 males aged between 28 and 46 were considered. This represents a limitation of the study, and future developments will address this by increasing the size of the dataset, including females, and extending the age range.”
5,Ensure consistency in the formatting of equations and figures throughout the manuscript. Some sentences are overly complex and could be simplified for improved readability. For instance, in the abstract, "Such tools have different measurement capabilities as well as physical principles over which the measurement procedure is based" could be rephrased as "These tools vary in measurement capabilities and the physical principles underlying the measurement procedures."
Response 5: Thanks for the suggestion. The paper has been read carefully and corrections to simplify too long and complex sentences have been carried out. Furthermore, equations and figures have also been double checked and, where needed, corrected.
Reviewer 3 Report
Comments and Suggestions for Authors
The work presented deals with one of the biggest concerns of the current society: the autonomy and/or the validation of using low cost devices for professional analysis.
In general lines, the work is well formulated, planned and developed. However, I think the authors should think about the following points:
- Abstract: When reading it, is not easy to see clearly how the study was developed. This is probably because the contextualization uses too general terms, which make it difficult to focus in the specific topic. Also, there are abbreviations that, if you don’t read the whole paper, you cannot understand. I recommend to avoid these kind of acronyms and to expose in a more concrete way the methodology used.
- Introduction: Same as in the abstract, there are very general reflections and it is not contextualized in a clear and direct way, which make the reading a bit heavy. There is a sub-section where ideas are repeated from the previous part. I suggest to reformulate this section to be short, clear and structured, avoiding repetitions and removing comments that maybe are not actual now, like the COVID-19 situation, unless it is really necessary. It is better to focus in the search of low cost resources and home autonomy, and to state the aim of the work in a clear and concise form.
- Methodology: Although it is correct in general, it is needed to improve the definition of inclusion and exclusion criteria, and to describe the intervention protocol more clear, schematic and ordered.
- Discussion: The authors should take into account and develop the points I have mentioned before, making sure there is coherence between methodology, results and interpretation.
- Conclusions: They should be reformulated based in the results, giving a clear, direct and concise answer to the objectives established.
Author Response
We thank the reviewers for revising the work so carefully, making a great contribution to the work. The reviewed manuscript has now been uploaded, and you can find the corrections in red throughout the text.
The work presented deals with one of the biggest concerns of the current society: the autonomy and/or the validation of using low cost devices for professional analysis.
In general lines, the work is well formulated, planned and developed. However, I think the authors should think about the following points:
Abstract: When reading it, is not easy to see clearly how the study was developed. This is probably because the contextualization uses too general terms, which make it difficult to focus in the specific topic. Also, there are abbreviations that, if you don’t read the whole paper, you cannot understand. I recommend to avoid these kind of acronyms and to expose in a more concrete way the methodology used.
Response 1: Thanks to the comment, we improved the overall readability of the paper, and in particular of the abstract, trying to avoid the use of acronyms and abbreviations whenever possible and adopting a clearer approach.
Introduction: Same as in the abstract, there are very general reflections and it is not contextualized in a clear and direct way, which make the reading a bit heavy. There is a sub-section where ideas are repeated from the previous part. I suggest to reformulate this section to be short, clear and structured, avoiding repetitions and removing comments that maybe are not actual now, like the COVID-19 situation, unless it is really necessary. It is better to focus in the search of low cost resources and home autonomy, and to state the aim of the work in a clear and concise form.
Response 2: We thank the reviewer for the constructive feedback. Following the suggestions, we revised the Introduction to improve clarity and structure, removing repetitions and outdated references (e.g., COVID-19). The section now focuses on the use of low-cost and accessible resources for motor assessment, emphasizing their relevance in contexts with limited equipment availability. Finally, the aim of the study has been clearly and concisely stated at the end of the section.
Methodology: Although it is correct in general, it is needed to improve the definition of inclusion and exclusion criteria, and to describe the intervention protocol more clear, schematic and ordered.
Response 3: We agree with the reviewer’s comment. We have included the inclusion criteria in the Participants section by adding the following sentence: “ Inclusion criteria were: to be considered as an Active subject and to not have any osteoarticular injuries and/or functional limitations that could interfere with the motor task.” Moreover we have edited the manuscript to provide a clearer and more structured description of the inclusion and exclusion criteria, as well as a more schematic presentation of the intervention protocol.
Discussion: The authors should take into account and develop the points I have mentioned before, making sure there is coherence between methodology, results and interpretation.
Conclusions: They should be reformulated based in the results, giving a clear, direct and concise answer to the objectives established.
Responses 4-5: Thank you for your insightful comment about Discussion and Conclusions. We have reformulated the Discussion session in order to make the text clearer and concise, avoiding reference to the Covid-19 (as you can read in the sentence “In recent years, there has been a growing tendency to adopt remote or hybrid assessment strategies. These approaches have spread due to their logistical simplicity and cost-effectiveness.”) and focusing on the point you stressed with the previous comment. We reformulated some sentences to avoid repetition and make the text less heavy to read (e.g. “In such contexts, it is common for different tools to be used throughout the year or training cycle, either due to changes in available equipment or the rotation of operators responsible for administering the assessments. For example, once the magnitude to be measured has been established, it may be useful to use cameras where the athlete cannot wear instrumentation, or wearable inertial systems where cameras cannot be used.”..........“Low-cost video-based systems often face limitations in real-world or non-elite contexts, being highly affected by factors such as frame rate, occlusions, camera angle, and image quality. The data processing approach proposed in this study offers a way to normalize vertical jump measurements, enabling a more reliable estimation of power output. This allows operators to obtain comparable and meaningful performance metrics even in decentralized or amateur settings, while maintaining consistency with established reference standards.”).
Moreover, we have also reformulated the conclusion to give a clearer answer to the objective established.
Round 2
Reviewer 1 Report
Comments and Suggestions for Authors
The authors have addressed my previous comments and concerns sufficiently, and I appreciate their detailed responses.